# Fatal Attraction: *Argiope* Spiders Lure Male *Hemileuca* Moth Prey with the Promise of Sex

**DOI:** 10.3390/insects15010053

**Published:** 2024-01-12

**Authors:** Andrew D. Warren, Paul M. Severns

**Affiliations:** 1McGuire Center for Lepidoptera and Biodiversity, Research Associate, Florida Museum of Natural History, University of Florida, 3215 Hull Rd., UF Cultural Plaza, Gainesville, FL 32611-2710, USA; awhesp@gmail.com; 2Department of Plant Pathology, University of Georgia, 2315 Miller Plant Sciences, Athens, GA 30602, USA

**Keywords:** predator–prey, coevolution, pheromone lure, chemical ecology, mate location

## Abstract

**Simple Summary:**

Predators and prey have direct interactions that influence their short-term behaviors, including resource allocation and strategies for moving through habitats. However, the presently observed behaviors are the products of coevolutionary interactions, posited to be a history of measures and countermeasures between the predator and prey. We found that *Argiope* (orb-weaver) spiders in the continental USA appear to use a pheromone lure that mimics the mating pheromone of the day-flying *Hemileuca* moth (buck moth) to entice male moths into their webs. We found evidence that different phylogenetic groups of *Hemileuca* moths respond to the *Argiope* pheromone lure with a broad range of responses, ranging from indifferent to acutely strongly attracted, suggesting a coevolutionary history of predator–prey countermeasures. One of these countermeasures may be the potential evolution of moth developmental timing (adult emergence) to avoid *Argiope* predation in areas where the ranges of the moths and spiders overlap.

**Abstract:**

Predator–prey coevolution, particularly chemo-ecological arms races, is challenging to study as it requires the integration of behavioral, chemical ecology, and phylogenetic studies in an amenable system. Moths of the genus *Hemileuca* (Saturniidae) are colorful, diurnal, and fast and often fly well above the vegetation canopy layer. However, several *Hemileuca* species have been reported as being captured in spider webs, specifically *Argiope* species (Araneidae). Female *Hemileuca* are known to produce mating pheromones and spiders of the Araneidae family are known to use pheromone lures to attract lepidopteran prey. We presented primarily female *Argiope aurantia*, which are attractive to male *Anisota pellucida* (Saturniidae), to different populations of *Hemileuca* species across the southern and western United States to categorize the homing response strength of different species of male *Hemileuca*. When we mapped these *Argiope* lure attraction strength categories onto the most recently published *Hemileuca* phylogeny, the behavioral patterns suggested a potential co-evolutionary arms race between predators and prey. Males of *Hemileuca maia*, *H. grotei*, and *H. nevadensis* (all in the same clade) appeared to have no attraction to *A. aurantia*, while *H. magnifica* and *H. hera* (within a different, separate clade) appeared to be strongly attracted to *A. aurantia*, but *H. nuttalli* (also within the *H. hera* and *H. magnifica* clade) displayed no attraction. Furthermore, *Hemileuca eglanterina* (yet a different clade) displayed strong, weak, and no attraction to *A. aurantia*, depending on the population. These apparent clade partitioning patterns of *Argiope* lure effectiveness and within-species variation in *Hemileuca* lure responses suggest a predator–prey coevolutionary history of measures and countermeasures.

## 1. Introduction

Coevolution, the reciprocal adaptation or counter-adaptation between interacting species, can drive organism diversification, population differentiation, and ecological specialization [1,2,3]. Initially conceived as an explanation for the tradeoffs between host plant specialization and host plant defenses [1], coevolution may also influence predator–prey interactions through repeated countermeasures of predator tactics and prey responses [4,5,6,7,8]. Although reciprocal behavioral adaptation by predators and prey may be rapid and relatively short-lived when compared with speciation rates [9], the signatures of predator–prey coevolution occur throughout nature, appearing as highly derived and specialized behavioral interactions. Studies of coevolutionary predator–prey behavioral interactions can only represent brief snapshots in evolutionary time. However, these studies imply that predator–prey systems with highly specialized interactions have most likely evolved to their present states over successive generations of predator specialization and prey countermeasures [10,11].

Much of the evidence for predator–prey coevolution arises from the study of chemo-ecological interactions. Along one axis of chemo-ecological interactions, arms races between toxic prey and toxin-resistant predators appear to generate a landscape mosaic of semi-independently evolving predator–prey populations that vary in their degree of toxicity, resistance, and predation/avoidance behaviors [12]. Over many generations, the landscape mosaic of different selective pressures and species’ interactions can diversify predator tactics and prey responses, potentially even facilitating the radiation of Müllerian and Batesian mimicry systems [13,14]. Along another axis of chemo-ecological evolutionary interactions, olfactory cues (scents and lures) emerge as potentially prominent drivers in predator–prey coevolution [11]. For species interactions that are driven by chemical cues, the prey species of ambush predators appear capable of detecting and modifying their behavior to reduce predation risk or avoid the predator altogether, while naïve prey species, or those with a new evolutionary history with a predator, are insensitive to predator chemical cues and experience a higher predation rate [15,16]. Such systems set up spatiotemporal metapopulation dynamics of predator–prey colonization and extinctions that can be labile and modified over evolutionary time scales.

Spiders are well-known for their use of chemical attractants to bait prey to their location [17]. As spiders tend to be mostly sessile, central-place foragers [17,18,19], the deployment of volatile compounds to bait and concentrate potential prey increases the chances of successful capture, compared to a passive approach of simple web placement. In one of the most thoroughly studied of these chemical lure systems, large immature and adult female bolas spiders of the genera *Mastaphora*, *Cladomelea*, and *Ordgarius* produce a palette of sex pheromone mimics to attract male moth prey. When baited into the spider’s capture range, the bolas spider lassos the moth with a highly modified, single-stranded web [20,21,22,23]. Juvenile bolas spiders of both sexes and adult male bolas spiders, on the other hand, emit pheromones to attract moth flies (Psychodidae), which are captured by the spiders at close range with their forelegs [24,25]. The secondary roles of spider pheromone volatiles appear to be diverse and widespread [17,26], suggesting the potential for predator–prey coevolutionary geographic mosaics of population and taxonomic differentiation similar to those observed in other chemo-ecological systems (e.g., garter snakes and their toxic amphibian prey [12]).

The degree to which pheromone lure systems have become specialized for prey attraction and how coevolution may be shaping these interactions is not well understood [26]. In North America, there are anecdotal accounts detailing the capture of large (8 to 10 cm), day-flying, male *Hemileuca* species (buck moths) (Saturniidae: Hemileucinae) in the webs of *Argiope* species (Araneidae) [27,28,29]. Male *Hemileuca* are vagile, strong-flying moths that typically fly well above vegetation canopies in grassland and shrub-dominated habitats [27,29,30]. However, the webs of *Argiope* in moth habitats are spun between plants, often within 1.5 m of the ground, where male *Hemileuca* rarely fly except to court females (adult *Hemileuca* have incomplete digestive systems so they do not seek food resources). Unmated female *Hemileuca* species emit a mixture of volatile pheromones that aid in mate location [31,32,33,34,35]. These pheromones trigger circular locating flights (often 500 m to 1 km in diameter) by males, in which, when the pheromone becomes concentrated enough, they then search up a pheromone gradient (~100 m) to precisely locate the calling female [27,29,30,35]. Observations of *Argiope* web locations and the capture of strictly male *Hemileuca* moths strongly suggest that female *Argiope* use a chemical lure/pheromone trap to bait male *Hemileuca* prey.

Chemically variable pheromones are emitted within and between different *Hemileuca* species, and they appear to be relatively diverse in mixture composition as well as in stereochemical structure [31,32,33,34,35]. Each *Hemileuca* species appears to have a potentially unique suite and mixture of pheromones, but even different populations of the same species can vary in their pheromone profiles [31,32,35]. Due to the standing diversity and variation of *Hemileuca* pheromones, it is possible that *Hemileuca*–*Argiope* predator–prey relationships are shaped by coevolutionary chemo-ecological interactions that vary over space and time—a predator–prey arms race. To evaluate this possibility, we characterized the relative attractiveness of female *Argiope* spiders regarding male *Hemileuca* homing for different *Hemileuca* species across North America and overlaid these interactions on the most recent and complete published *Hemileuca* phylogeny. A coevolutionary predator–prey arms race should generate a mosaic of species–species interactions across the phylogeny where the pheromone lure is effective and is not countered by the prey, along with other instances where the pheromone lure is ineffective. In a phylogenetic framework, evidence of coevolutionary interactions between *Hemileuca* and *Argiope* could be manifested as clade-specific patterns of lure effectiveness and/or a wide range of lure responses within a clade or a single *Hemileuca* species.

## 2. Materials and Methods

### 2.1. Study Species

*Argiope* are large (~3–7 cm), colorful, orb-weaving spiders (Araneidae), which are distributed throughout the world. *Argiope aurantia,* commonly known as the black-and-yellow garden spider, is frequently observed in gardens, fields, and along lake edges across the United States, although it is less common in the western Great Plains, the Rocky Mountain region, and in the western deserts [36]. Adults of *A. aurantia* mature in late summer through autumn in most of its range [37], although they mature as early as June in Florida (Warren pers. obs.). *Argiope* are general insect feeders and accept a very wide range of prey items. Mature *A. aurantia* females may live as long as 6 months (Warren pers. obs.) Throughout North America, the ranges of *A. aurantia* and *Hemileuca* (see below) broadly overlap.

*Hemileuca* (buck moths) are relatively large (8–10 cm), colorful, day-flying Saturniidae (silk moths) that are restricted to North America and Mexico [29]. Like most adult saturniid moths, *Hemileuca* adults lack functional feeding mouthparts, and their digestive systems are incomplete [27,29]; thus, the adults do not feed and are short-lived [27,29,30]. *Hemileuca* mate location, at the broader landscape level, is accomplished through volatile sex pheromones. These pheromones can occur as a dominant volatile or as a mixture of multiple volatiles [31,32,33,34,35,38]. *Hemileuca* pheromones are known to be species-specific, but they may be cross-compatible as the pheromone components of one species may attract the males of another [34,39]. During mate location, male *Hemileuca* often fly in large circles through a habitat where females are likely to occur. When a pheromone plume is detected, the flight paths become smaller in diameter, enabling the males to geospatially map where the pheromone concentrations are greatest, ultimately following the pheromone concentration gradient to the emitting female [27,40].

### 2.2. Evaluating Argiope Lure Attraction to Male Hemileuca

Mature or last-instar immature female *Agriope aurantia* (hereafter *Argiope*) that were observed to attract diurnal male *Anisota pellucida* (Saturniidae) (up to 20 individual *A. pellucida* moths each day; see Appendix A) were gathered from the vicinity of Gainesville, FL, USA, and kept in 0.75 m × 0.75 m rectangular mesh collapsible cages. The captive *Argiope* were fed crickets purchased from PetSmart every 2–3 days, a diet occasionally supplemented with wild-harvested grasshoppers; at no point were the spiders offered *Hemileuca* prey. Caged *Argiope* were kept in a manner that would maximize their health and welfare throughout the duration of the experiment, including regular misting with water and exposure to sunlight. During field assays from 2016–2018, immature female *Argiope* were also gathered and presented at field sites to determine which instars attract moths and it was determined that only mature or last-instar *Argiope* appeared capable of emitting volatile *Hemileuca* lures.

At the study sites, side-by-side cages (one cage containing a calling *Argiope* and one cage without a spider) were deployed ~5 m apart from each other, from the late morning to mid-afternoon when *Hemileuca* were actively flying. Cages were placed in habitats with low vegetation and ample exposure, such as ridge tops, hilltops, and elevated roads, so that incoming male *Hemileuca* moths could be clearly observed and to enhance the dispersal of emitted pheromones. Because spider webs themselves may visually attract prey [41,42,43], we presented *Argiope* in their cages to counter any signal based on the webs and to standardize their presentation to *Hemileuca*.

We developed a relatively straightforward set of behavioral criteria for categorizing the degree of attraction to cages (with and without *Argiope*) by searching male *Hemileuca* moths within 10 m of the cages. Although this 10-meter distance may or may not be the male moths’ true perceptual range, there was a clear behavioral response to *Argiope* spiders at this distance that was straightforward to identify and record for individual males. To be categorized as being strongly attracted to *Argiope*, *Hemileuca* males had to deviate from their flight path, either contact the ground immediately in front of the cage or contact the cage itself, and remain near the cage location (<2.5 m) for more than two minutes. We considered a deviation from the original flight path and movement towards the cage without stopping or contacting the cage as “weak attraction”. No attraction was assigned to a male that did not deviate from its flight path (within a 10-meter radius of *Argiope*) and orient towards *Argiope*. For each male that came within 10 m of the experimental *Argiope,* we recorded whether it was strongly attracted, weakly attracted, or not attracted to the “calling *Argiope*”.

In our behavioral assays, indexing moth attraction was important, and we were also concerned with other potential sources of attraction besides the *Argiope*. We directly assessed potential alternative sources of attraction/repulsion/indifference prior to formal data collection. First, the cages alone may either attract or repel *Hemileuca*, due to unknown and uncharacterized properties. After placing unoccupied cages near *Argiope*-occupied cages and unoccupied cages in isolation, it was immediately apparent that the unoccupied cages were not attractive to male *Hemileuca*. Although the cages did not appear to be repulsive to male *Hemileuca* (e.g., turning away from the cages once a perceptual range threshold was crossed), it is possible that the cages produce repulsive signals at close distances that are overwhelmed by the *Argiope* volatile. Second, not all *Argiope* appeared to emit the lure at all times, but some individuals reliably called for longer periods of time (daily, for weeks) while others only called on some days. Therefore, we frequently presented multiple *Argiope aurantia* (up to 12 individuals in separate cages), to ensure a high likelihood that at least one individual was emitting. Occasionally, we also evaluated two additional *Argiope* species (*A. florida* and *A. trifasciata*). Although we omit these species from formal statistical analysis due to limited sampling and experimental effort, we consider them as potentially relevant biological observations. Last, after we established that a caged *Argiope* putatively called and attracted a number of male *Hemileuca*, we moved the cage to another location >5 m away as a manipulation test of lure attractiveness. In these instances, some males remained for 5 to 10 min at the location where the cage had previously been positioned, but most males reoriented to the new location of the calling spider within 1 min. We also noted individual variations in the *Argiope* attractiveness of lures that varied over space and time. Multiple caged *Argiope* may call simultaneously at the same site but there appeared to be obvious individual differences in lure effectiveness (Appendix A). In other instances, an individual spider would be an effective caller one day but not the next. Together, these observations strongly suggest that relative lure attractiveness was estimated with our methodology and that the lure-emitting system is under some degree of individual control.

### 2.3. Lure Attraction and Overlay on a Published Hemileuca Phylogeny

We compared the percentage of *Hemileuca* moths displaying evidence of strong attraction to the caged *Argiope* to the percentage of those that did not display strong attraction (with weak and no-attraction combined) via a two-tailed Z-proportions test (α < 0.05) to assign each population a categorical lure response. In opting for a proportions test, we assumed that the number of spiders deployed at any site on any date would not impact the homing behavior to a degree that would overcome the broader (more conservative) properties of the proportions test based on a Z-statistic. For *Hemileuca* species containing more than one study population, if >80% of the aggregate populations were categorized as strongly attracted, we designated that species with a strong lure attraction response (the same for the no-lure response category). If the percentage of populations did not statistically differ from a 50% ratio of no attraction (and weak attraction) to strong attraction, we considered that particular *Hemileuca* taxon to have a mixed lure response. We then mapped these three lure attraction categories, namely, strong attraction, no attraction, and mixed attraction, for each *Hemileuca* species with behavioral lure assays (*H. eglanterina*, *H. grotei*, *H. hera*, *H. magnifica*, *H. maia*, *H. nevadensis*, and *H. nuttalli*) (Appendix A) onto the most recent published *Hemileuca* phylogeny, as presented by Rubinoff and Sperling [36]. The phylogeny we selected was the maximum likelihood tree produced from combined cytochrome oxidase I (COI—mtDNA) and elongation factor 1 alpha (EF1α—nuclear DNA) ([36] Figure 6) because that tree was constructed from the greatest amount of sequence information. By overlaying these assayed *Argiope* lure categories onto the published *Hemileuca* phylogeny, we could evaluate whether there may be a phylogenetic signal associated with *Argiope* lure attraction.

## 3. Results

Seven *Hemileuca* species, namely, *H. eglanterina*, *H. grotei*, *H. hera*, *H. magnifica*, *H. maia*, *H. nevadensis* and *H. nuttalli*, were assayed for *Argiope* lure attraction across the western, central, and southern USA from populations in the states of New Mexico, Utah, Colorado, Wyoming, Texas, Georgia, and Florida (Appendix A). A total of 33 populations were presented with caged *Argiope* for an aggregate observational time of ~127 h (Appendix A).

When we overlaid the species categories of *Argiope* lure attraction on the published *Hemileuca* phylogeny, there appeared to be evidence of phylogenetic association related to lure effectiveness. All sampled populations of *Hemileuca grotei*, *H. maia*, and *H. nevadensis* consistently showed no homing responses by any individual male moths to caged *Argiope* (Appendix A); these taxa occurred in clades 1 and 2 (Figure 1). Likewise, strong lure attractiveness appeared in the *Hemileuca* taxa associated with clade 3 (*H. hera* and *H. magnifica*), but this clade also contained species without any evidence of attraction (*H. nuttalli*), along with one species containing populations that were either strongly attracted, weakly attracted, or showed no evidence of attraction (*H. eglanterina*) (Figure 1, Appendix A).

For *H. hera* and *H. magnifica*, there was one population of each species where *Argiope* lures appeared to be weakly or non-attractive to most male moths (Appendix A). It is likely that in these instances the *Argiope* had ceased lure emission as moths were attracted to the same *Argiope* on a later date or later that day in a close-proximity subpopulation (Appendix A). We also noted that *Arigiope florida* and *A. trifasciata* did not appear to emit a lure that elicited a homing response from male *H. grotei*, *H. maia*, and *H. nevadensis*. However, these three *Hemileuca* species were not documented as being attracted to *A. aurantia* either (Figure 1), and the presentation of *A. florida* and *A. trifasciata* was an infrequent, late-season occurrence (Appendix A).

## 4. Discussion

In our research, there was evidence of varied *Argiope* lure effectiveness, suggesting that a chemical coevolutionary arms race may be occurring between *Argiope aurantia* and moths of the genus *Hemileuca*. First, although we performed lure behavioral bioassays on seven *Hemileuca* species occurring throughout the western, central, and southern USA, we have representatives from 3 of the 4 primary *Hemileuca* clades defined by Rubinoff and Sperling [36]. We observed a variation in apparent *Argiope* lure effectiveness among taxa between clades and there may be a phylogenetic association with whether or not the *Argiope* lure attracts male *Hemileuca* (Figure 1). None of the clades were fully represented in our behavioral assays, but members of clades 1 and 2 (*H. grotei*, *H. maia*, and *H. nevadensis*) showed no evidence of lure attraction in the species and populations evaluated (Figure 1, Appendix A). Not a single instance of weak attraction was observed in any *H. grotei, H. maia*, or *H. nevadensis* individual, suggesting that although the spiders were likely emitting pheromones, those volatile chemicals were not capable of inducing a homing response towards the *Argiope*. Counter to our results, Horton [28] reported the finding of *Hemileuca lucina*, which is closely related to *H. maia* and *H. nevadensis* (though not included in the phylogeny of Rubinoff and Sperling [36]) in the webs of *Argiope*. There are also anecdotal reports of male *Hemileuca clio* (placed within clade 2 as defined by Rubinoff and Sperling [36]), which is distantly related to *H. maia* and *H. nevadensis*, being attracted to *Argiope* in Arizona which was verbally communicated to the first author by M. Collins and S. McElfresh. Thus, the distribution and intensity of *Argiope* lure attractiveness across clade 2 appears to be more variable and is perhaps even biogeographically partitioned in a way not revealed by our study. The clade-specific sorting of traits may emerge as a consequence of a relatively long-standing coevolutionary history between *Argiope* and *Hemileuca* (as opposed to a recently evolved interaction), but we cannot validate this evolutionary history with the data we present.

Although the taxa in *Hemileuca* clades 1 and 2 are sparsely represented, the *Hemileuca* spp. in clade 3 (Figure 1) were more thoroughly assayed for *Argiope* lure responses. The patterns of *Argiope* lure responses by male *Hemileuca* in clade 3 suggest the potential for an ongoing evolutionary chemical arms race between predator and prey. We found clear evidence of strong lure attraction to *Argiope* by *H. hera* and *H. magnifica*. In some populations, *Argiope* attracted males in the high hundreds to their cages (Appendix A). Yet, within this same clade, there was no evidence of attraction, even weak attraction, to *Argiope* by any individuals of *H. nuttalli* from multiple different populations (Appendix A). This suggests the potential finding that either *Argiope* pheromones may have recently evolved to lure *H. hera* and *H. magnifica*, that *H. nuttalli* has recently evolved with an indifference to the lures, or that the lures were potentially never effective with *H. nuttalli*. We do not yet have the resolved information that would enable a strongly supported answer to these three scenarios, but the patterns of strong lure effectiveness and lure ineffectiveness for baiting *Hemileuca* prey is another expected consequence of predator–prey coevolutionary arms races [1,2,3,4,5,6].

Perhaps the most interesting and biologically revealing patterns of predator–prey coevolutionary responses in lure attraction reside with the species *H. eglanterina*. We had evidence that the *Argiope* lure spans the range of male moths’ homing responses between *H. eglanterina* populations, including no attraction, strong attraction, and weak attraction to the caged spiders (Appendix A). Interestingly, *H. eglanterina* is known to have a mixture of female moth pheromone volatile constituents that, when mixed in certain proportions, are effective at attracting male *H. eglanterina* in some populations and not in others [32,34]. It is possible that the pheromone constituent diversity in *H. eglanterina* either predisposes it to *Argiope* lure attraction or to indifference, or that perhaps the pheromone mixtures are evolving in response to *Argiope* lures. Although we do not know the *Argiope* lure’s chemical composition, it is also possible that they too may be capable of producing different volatile mixtures since bolas spiders, which are also Araneidae, are known to do this to target specific prey [23,24,25].

*Argiope* predation on male *Hemileuca* in nature remains anecdotal, unquantified, and infrequently reported in the literature [27,28,29]. There are two speculative but potentially revealing observations we present, based on the qualitative knowledge gained during our study. First, while *Argiope* occur in the majority of the *Hemileuca* habitats assayed during this study, and their ranges in North America broadly overlap [29,37], locally, *Argiope* usually mature (the calling life stage of *Argiope*) following the main flights of summer-flying *Hemileuca* (*H. eglanterina, H. nuttalli, H. hera,* and *H. magnifica*), temporally separating predator from prey. Our experimental introduction of mature female *Argiope* into *Hemileuca* populations during their peak flight represents a scenario that is unlikely to occur naturally in these populations. Given the intensity of the attraction of male *H. hera* and *H. magnifica* to mature, calling *Argiope*, it is possible that the chemical arms race between these taxa has helped shape the mid-summer flight times of these species, as opposed to the late fall flight times of species in clades 1 and 2. In clades 1 and 2, these taxa tend to have a long pupal diapause stage that extends into the late fall and early winter months [29], wherein *Argiope* may or may not overlap with *Hemileuca* flights. While the following proposal is speculative, and while there are birds and large dragonflies (Aeshnidae) that are adult moth predators, it is possible that either the flight times of *Hemileuca* coincidentally do not overlap with the *Argiope* life stages that are capable of predating adult *Hemileuca* or that *Hemileuca* development has evolved to avoid *Argiope* predation. An example of this potentially resides in the *H. maia* group, where the adult moths emerge from late October through early January (depending on latitude), flying after adult *Argiope* have died. Second, the *Argiope* lure may not be restricted to *Hemileuca* nor to *Argiope aurantia*, as we have evidence of strong lure attraction to another saturniid moth genus, *Anisota* (Saturniidae: Ceratocampinae), and evidence that other *Argiope* species likely produce a moth pheromone lure as well (Appendix A). Third, there may be asymmetric sexual selection for sensitivity to the *Argiope* lure as we never recorded 100% of the observed males homing to caged *Argiope*, nor were any female *Hemileuca* observed to be baited by the spiders (Appendix A). The skewed sex ratio suggests that it is likely that the predator–prey coevolutionary interactions differ for male and female moths. Finally, a mating pheromone of *Argiope bruennichi* has been published [44] but it remains to be tested whether the *Argiope* mating pheromone can also bind and elicit the homing response in *Hemileuca* or if the *Argiope* lure is produced independently of the spider mating pheromone.

## Figures and Tables

**Figure 1 insects-15-00053-f001:**
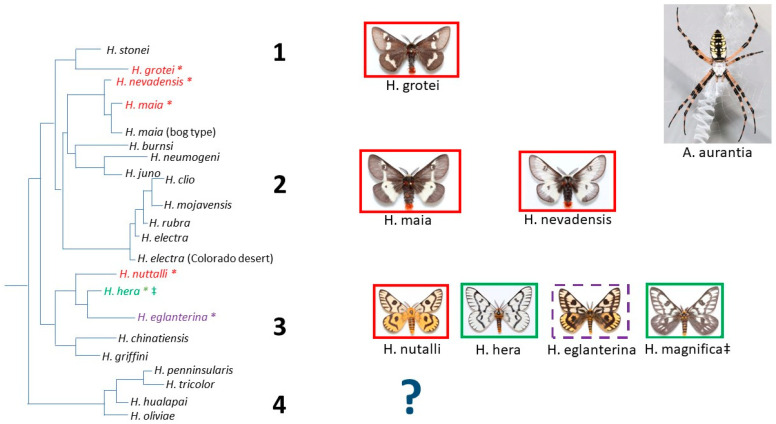
*Hemileuca* phylogeny, as presented by Rubinoff and Sperling [36] (Figure 6), including the proposed clades (1–4) and position of the *Hemileuca* taxa (pictured) evaluated for male attraction to caged *Argiope aurantia*. Boxes around each *Hemileuca* taxon signify whether there was no attraction (red box, and red taxon name with an *), strong attraction (green box with a green taxon name and an *), or a mixture of populations with no or strong attraction (purple hashed border with a purple taxon and an *) to caged *A. aurantia* (Appendix A). We did not evaluate *Hemileuca* attraction from any members of clade 4 and *H. magnifica* (‡) was originally considered a subspecies of *H. hera*, so we assume that its position would be in clade 2 with *H. hera*.

## Data Availability

Behavioral assay data, site locations, dates, and proportion Z-test results are presented in the Appendix A.

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
