# Peer review of "Fatal Attraction: Argiope Spiders Lure Male Hemileuca Moth Prey with the Promise of Sex"

_insects, 2024, doi:10.3390/insects15010053_

Round 1
Reviewer 1 Report
Comments and Suggestions for Authors
Author Response
Please find the original comments and suggestions from reviewer 1 below and our response to those suggestions including modification to the manuscript for the revised version.
This paper is extremely well written and well organized. The study has been carefully defined,
trimming out parts that are anecdotal or with minimal observations, so only using the primary
data that appear to be supported by many observations. The authors are careful to distinguish
between fact and speculation, although they appropriately offer some tantalizing hypotheses.
The paper is free of misspellings and grammatical errors. The style of writing is engaging and
effective, revealing the enthusiasm of the authors.
I have the following suggestions:
Line 61, add an umlaut to Müllerian or spell the name as Muellerian.
Thank you for catching this we have made the change to the correct German spelling.
Line 62, add the word “mimicry” after Batesian.
Thank you for catching this omission we have added mimicry as suggested.
Line 90, Araneidae is misspelled.
Thank you for catching this typo, it is now spelled correctly.
Line 129, add Mexico after North America. Yes, parts of Mexico are biogeographically part of
North America, but some Hemileuca fly in Chiapas, Baja California, etc.
Thank you for catching this inaccurate distribution. We both know this but we failed to recognize the mistake!
Line 129, add the word most: Like most adult saturniid moths…. On two occasions in the 1880s.
Citheronia regalis has been reported to feed at sugary bait (Ferguson MONA 1971: 33) and four genera of Attacini are recorded to drink water.
Thank you for reminding us that there are observations of Saturniids drinking. We have modified the sentence to read “Like most adult saturniid moths..”
Lines 319-322. To suggest that some Hemileuca have shifted to a two-year life cycle in response to predation by Argiope is quite a stretch. Some Coloradia (related to Hemileuca) have a two-year life cycle, so one might wonder if Argiope might attract males of Coloradia at night. But to return to the main point here: it seems more likely that the two-year life cycle evolved because there is not enough time to complete it all in a single season from spring to fall in some areas where freezing weather can persist into late spring and occur already in early fall. Some montane butterflies have a two-year life cycle, but I do not know the hypotheses for those (but the senior author would be very familiar with these cases). In any case, I do not think predation pressure from Argiope could be significant enough to cause a shift from a one- to two-year life cycle.
I am inclined to suggest that perhaps the maia-group has not co-evolved with the Argiope because those moths fly so late in the year (often December and even January) after the spiders for that year have died.
I would like to see a few comments pertaining to other predators of Hemileuca. Surely birds are significant. Have so many species of Hemileuca evolved to be diurnal to avoid bat predation? I realize these questions would clutter the main point of the paper, but some readers may be left thinking that Argiope are the main predators of Hemileuca moths, which I doubt.
Looking at the internet, it appears that some Europeans have identified the pheromone in a species of Argiope in Europe. The chemical components of the sex pheromones in Hemileuca have been reported in some of the papers that these authors cited. I am not suggesting any biochemical analyses need to be done for this study, but it would be desirable to note if the published pheromones of the spider and the moths have any of the same active groups.
In summary, this is an excellent study very worthy of publication by this journal.
We were unclear in our manuscript and have revised the Discussion to be more consistent with our original intent and integrated the reviewer’s suggestions into the final paragraph of the manuscript. There was clearly some miscommunication on our part as we were not proposing a two year life cycle for Hemileuca but rather proposing that their adult flight times may have evolved to not overlap with Argiope. We used the H. maia group as a potential example as the reviewer suggested and have made clear that there are likely other predators of Hemileuca than Argiope spiders. We have revised the final paragraph of the manuscript and indicated our changes in the marked-up version. The final paragraph now reads,
“Argiope predation on male Hemileuca in nature remains anecdotal, unquantified and infrequently reported in the literature [27-29]. There are two speculative but potentially revealing observations we present based on the qualitative knowledge gained during our study. First, while Argiope occur in the majority of Hemileuca habitats assayed during this study and their ranges in North America broadly overlap [29,37], locally Argiope usually mature (the calling life stage of Argiope) following the main flights of summer-flying Hemileuca (H. eglanterina, H. nuttalli, H. hera, H. magnifica), temporally separating predator from prey. Our experimental introduction of mature female Argiope into Hemileuca populations during their peak flight represents a scenario unlikely to occur naturally in these populations. Given the intensity of the attraction of male H. hera and H. magnifica to mature, calling Argiope, it is possible that the chemical arms race between these taxa has helped shape the mid-summer flight time of these species, as opposed to the late fall flight time of species in clades 1 and 2. In clades 1 and 2, these taxa tend to have a long pupal diapause stage that extends into the late fall and early winter months [29], where Argiope may or may not overlap with Hemileuca flights. While speculative, and there are bird and large dragonflies of the Aeshnidae as adult moth predators, it is possible that the flight times of Hemileuca either coincidentally do not overlap with the Argiope life stages capable of predating adult Hemileuca or that Hemileuca development has evolved to avoid Argiope predation. An example of this potentially resides in the H. maia group where the adult moths emerge from late October through early January (depending on latitude), flying after adult Argiope have died. Second, the Argiope lure may not be restricted to Hemileuca nor to Argiope aurantia as we have evidence of strong lure attraction to another saturniid moth genus Anisota (Saturniidae: Ceratocampinae) and evidence that other Argiope species likely produce a moth pheromone lure as well (Supplemental Table 1). Third, there may be asymmetric sexual selection for sensitivity to the Argiope lure as we never recorded 100% of the observed males homing to caged Argiope and nor were any female Hemileuca observed to be baited by the spiders (Supplemental Table 1). The skewed sex ratio suggests it is likely that the predator-prey coevolutionary interactions differ for male and female moths. Last, a mating pheromone of Argiope bruennichi has been published [44] but it remains to be tested whether the Argiope mating pheromone can also bind and elicit the homing response in Hemileuca or if the Argiope lure is produced independently of the spider mating pheromone.”
Reviewer 2 Report
Comments and Suggestions for Authors
General comments
Coevolution has intrigued scientists for decades ever since the publications of Ehrllch and Raven. And, as noted by the authors, many coevolutionary studies have featured chemo or chemosensory drivers. In the current submission, the authors focused on possible coevolution between deceptive odours from garden spiders that attract buck moths. The paper is well written and easy to follow and the authors clearly indicate where their conclusions require further support. An impressive amount of work suggests some interesting patterns have evolved within this system.
Specific comments
- The introduction is well written and well populated with appropriate references.
2) In the Introduction, the criteria for concluding a coevolutionary mosaic seem somewhat vague. The authors need to be more specific regarding the critical criteria for concluding such. For example, there may be alternate explanations for a mosaic-of-response intensity. Similarly what data are critical to showing that the patterns observed are due to a reciprocal adaptation, a key feature of this phenomenon?
3) Field methodology was well explained.
4) Line 193 - “ Last, after we established that a caged Argiope effectively called …”. How do did the authors confirm the act of calling or did they mean putatively or apparently called?
5) Cages apparently varied in the number of spiders present, which suggests uncontrolled variation in intensity of putative pheromone release. In addition, there were different sites assessed across different dates. How were these random elements captured in a proportions test?
6) Some estimate of potential impact of Argiope on Hemileuca would be useful to buttress the coevolutionary story the authors are attempting to weave. (see (7) below)
7) The authors should consider other possible reasons for the results observed, for example, region-specific impacts on pheromone calling efficiency since they are also likely observing asymmetric sexual selection on males and females.
Author Response
Please find the original comments and suggestions from reviewer 1 below and our response to those suggestions including modification to the manuscript for the revised version.
Reviewer 2
Coevolution has intrigued scientists for decades ever since the publications of Ehrllch and Raven. And, as noted by the authors, many coevolutionary studies have featured chemo or chemosensory drivers. In the current submission, the authors focused on possible coevolution between deceptive odours from garden spiders that attract buck moths. The paper is well written and easy to follow and the authors clearly indicate where their conclusions require further support. An impressive amount of work suggests some interesting patterns have evolved within this system.
Specific comments
- The introduction is well written and well populated with appropriate references.
2) In the Introduction, the criteria for concluding a coevolutionary mosaic seem somewhat vague. The authors need to be more specific regarding the critical criteria for concluding such. For example, there may be alternate explanations for a mosaic-of-response intensity. Similarly what data are critical to showing that the patterns observed are due to a reciprocal adaptation, a key feature of this phenomenon?
We added a sentence to the end of the last paragraph of the Introduction section which now reads, “In a phylogenetic framework, evidence of coevolutionary interactions between Hemileuca and Argiope could be manifested as clade specific patterns of lure effectiveness and/or a wide range of lure responses within a clade or a single Hemileuca species.”
Because we are at the very beginning of this inquiry, tracking individual populations of Argiope and Hemileuca are not yet possible to unambiguously demonstrate through empirical observation and change of predator-prey measures and countermeasures.
3) Field methodology was well explained.
4) Line 193 - “ Last, after we established that a caged Argiope effectively called …”. How do did the authors confirm the act of calling or did they mean putatively or apparently called?
Good point, thank you for this comment, we modified line 193 to “putatively called”.
5) Cages apparently varied in the number of spiders present, which suggests uncontrolled variation in intensity of putative pheromone release. In addition, there were different sites assessed across different dates. How were these random elements captured in a proportions test?
This is one of the weaknesses of working in a new system with somewhat unpredictable predator behavior. We do not have the Argiope pheromone that attracts Hemileuca and the Argiope and moth homing behaviors (at an individual level) appear to interact in a way that we do not understand. We attempted to explain the pertinent biology and behavior of the players in the system in lines 180-207, to indicate that there are many aspects we are unsure of and that are difficult to work with. We also could not track individual male Hemileuca to unambiguously determine differences in individual responses to lures. So, we made the decision to analyze only the proportion of Hemileuca homing in each of the homing categories, as this coarse approach to analysis was the most straightforward way to deal with the data (zeros were a problem in populations where Hemileuca moths appeared indifferent to Argiope). We modified the methods section to acknowledge that we make important assumptions when we ran the proportions test. This sentence reads, “In opting for a proportions test, we assumed that the number of spiders deployed at any site on any date would not impact the homing behavior to a degree that would overcome the broader (more conservative) properties of the proportions test based on a Z-statistic.”
6) Some estimate of potential impact of Argiope on Hemileuca would be useful to buttress the coevolutionary story the authors are attempting to weave. (see (7) below)
7) The authors should consider other possible reasons for the results observed, for example, region-specific impacts on pheromone calling efficiency since they are also likely observing asymmetric sexual selection on males and females.
Thank you for bringing to our attention asymmetric sexual selection. While we were focused on the predator prey interactions in a more general context, it is true that the coevolutionary implications of this proposed predator prey interaction are modified by the sex of the moth. To make this point, and admit that it is another interesting aspect of this system, we modified the last paragraph of the manuscript. The passage referring to this modification now reads,
“Third, there may be asymmetric sexual selection for sensitivity to the Argiope lure as we never recorded 100% of the observed males homing to caged Argiope and nor were any female Hemileuca observed to be baited by the spiders (Supplemental Table 1). The skewed sex ratio suggests it is likely that the predator-prey coevolutionary interactions differ for male and female moths.”
Round 2
Reviewer 2 Report
Comments and Suggestions for Authors
The authors have largely satisfied my concerns.